# Influence of Parallel Tubular Channel Angular Pressing (PTCAP) Processing on the Microstructure Evolution and Wear Characteristics of Copper and Brass Tubes

**DOI:** 10.3390/ma15092985

**Published:** 2022-04-20

**Authors:** Mohamed Ibrahim Abd El Aal, Hossam Hemdan El-Fahhar, Abdelkarim Yousif Mohamed, Elshafey Ahmed Gadallah

**Affiliations:** 1Mechanical Engineering Department, College of Engineering at Wadi Addawaser, Prince Sattam Bin Abdulaziz University, Wadi Addawaser 18734, Saudi Arabia; 2Mechanical Design & Production Department, Faculty of Engineering, Zagazig University, Zagazig 44519, Egypt; 3Mechanical Production Department, Faculty of Technology & Education, Suez University, Suez 43527, Egypt; hossam.alfaghaar@suezuniv.edu.eg (H.H.E.-F.); abdelkarim.mohamed@ind.suezuni.edu.eg (A.Y.M.); elshafey.gadallah@ind.suezuni.edu.eg (E.A.G.)

**Keywords:** parallel tubular channel angular pressing (PTCAP), copper and copper alloy tubes, hardness, microstructure, wear characteristics

## Abstract

The influence of the number of passes and the tube materials on the microstructural evolution, mechanical properties, and wear behavior of Cu and brass tubes after parallel tubular channel angular pressing (PTCAP) was investigated. The grain size decreased to final grain sizes of 138.6 nm and 142.7 nm, after PTCAP of the Cu and brass tubes was conducted in up to 4 and 2 passes, respectively. PTCAP contributes to obtaining an ultra-fine grain (UFG) microstructure, with a mixture of different grain sizes that conferred high hardness. The present results indicate the superior wear resistance of Cu and brass PTCAP tubes, relative to Cu and brass samples that were previously deformed by different severe plastic deformation (SPD) processes. The wear mechanism of the Cu tubes changed from delamination and cracks with a high degree of adhesive wear before PTCAP into a combination of adhesive and abrasive wear, with a decrease in the presence of oxygen content after the PTCAP procedure. The wear mechanism also changed from a combination of adhesive and abrasive mechanisms into abrasive ones with the absence of oxygen after the PTCAP of brass tubes.

## 1. Introduction

The effect of grain refinement using the severe plastic deformation (SPD) of metal and metal alloys in improving their mechanical and physical properties has motivated many scientific investigations. SPD comprises different techniques, especially equal channel angular pressing (ECAP) and high–pressure torsion (HPT) [1,2,3,4,5,6,7,8,9]. SPD methods are concerned with processing samples of different shapes, including bulk, cylindrical, and ring samples. Unfortunately, few previous works are concerned with producing fine-grain structured tubular components using SPD methods [10,11,12]. Previous approaches include the use of ECAP, spin extrusion (SE) [13], high–pressure tube twisting (HPTT) [14,15], and the accumulative spin–bonding (ASB) method based on accumulated roll–bonding (ARB) [16]. However, these techniques were highly expensive, it is difficult to set up their dies, they have limited application, and had some limitations such as low homogeneity of the produced microstructure.

Novel SPD methods and techniques have been used recently [17,18,19]; moreover, new SPD processes, known as tubular channel angular pressing (TCAP) and parallel tubular channel angular pressing (PTCAP) [20,21,22,23], have been proposed. However, PTCAP is considered better as it needs a lower load and leads to better strain homogeneity [24,25,26,27,28]. Copper and brass tubes are considered essential products that are used in different applications. Recently, the demand for high strength, hardness, superior wear, and corrosion resistance in Cu and brass tubes has increased.

The production of fine-grain-microstructure Al, Cu, and brass tubes using PTCAP has motivated various studies [21,24,25,26,27,29]. Recently, research by both Faraji et al. and M. Mesbah et al. [20,21] was concerned with the grain refinement of Cu tubes using PTCAP. The PTCAP process was influential in grain refinement, improving the Cu and brass tubes’ mechanical properties [20,21,25,26,27,28]. Unfortunately, the grain refinement that occurred after PTCAP was combined with a decrease in Cu and brass tube ductility. Interestingly, most of the analyses of previous works with the PTCAP of pure Cu and brass were performed using the same die angles of 120°–150° and a pressing rate of 5–10 mm/min. Recently, further investigations using a combination of smaller die angles with a slower pressing rate for producing a microstructure with a mixture of fine- and coarse-grained brass tubes of high hardness and wear resistance with good ductility were recently conducted [30]. However, using the same conditions in the PTCAP of Cu or other brass tube types has not been attempted until now. The presence of a microstructure with a mixture of fine and coarse grains can be considered a multi-modal situation that consists of a mixture of submicron and micron sizes; therefore, it can provide a considerably high level of strength and hardness while conserving a reasonable degree of ductility [31,32,33] relative to the SPD materials, with mono-modal ultrafine grains that increase strength and hardness without sacrificing ductility.

The influence of SPD processing on the wear characteristics of Cu and Cu alloys was the motivation behind other recent studies [34,35,36]. An ECAP with up to 8 passes over the Cu–0.1% Zr alloy effectively improved its wear properties [34]. This observation was further confirmed in the case of brass tubes processed by constrained groove pressing (CGP) [35]. Therefore, SPD processing enhanced the wear characteristics of copper and its alloys. Unfortunately, there are no previous studies concerned with investigating the influence of TPCAP on the wear characteristics of Cu and brass tubes.

The present research aims to: Investigate the influence of PTCAP on the microstructure evolution of Cu and brass tubes in terms of producing a microstructure with a mixture of fine and coarse grains.Study the influence of grain refinement after the PTCAP process on the mechanical properties of the Cu and brass tubes, mainly in terms of hardness and wear mass loss under different applied load and sliding distances.Investigate the influence of PTCAP on the wear mechanism and wear scars of the Cu and brass tubes.Confirm the wear mechanism results through a study of the wear surface morphology and analysis of the Cu and brass tubes.

## 2. Materials and Methods

### 2.1. Materials, Microstructure, and Hardness 

Pure Cu and brass tubes, with compositions as listed in Table 1 and an outer diameter, thickness, and length of 20, 2.5, and 80 mm, respectively, were used in the present work. The different tubes were annealed for 1 h at 700 °C in a vacuum furnace under argon gas. PTCAP was performed using an H13 tool steel die with parts including a mandrel, die, and punches, as shown in Figure 1. The die geometry parameters include channel angles φ_1_ = φ_2_ of 135°, angles of curvature ψ_1_ = ψ_2_ of 0°, and tube thickness K = R_2_ – R_1_ = 2.5 mm (Figure 1a). The PTCAP die parameters were selected based on a previous finite element study [29,37]. Low die angles have been recommended to introduce greater strain homogeneity and finer grain. Therefore, an angle of 135° was chosen to achieve UFG and high strain (deformation) homogeneity. PTCAP of the pure Cu and brass tubes was performed at room temperature (RT) using molybdenum disulfide (MoS_2_) as a lubricant [30].

The Cu and brass tubes were processed up to 4 times in 2 passes at a 2 mm/min speed using a 300 KN universal METRO COM model (10334, Garbagna, Novarese, Italy) (Figure 1b–d). The PTCAP process consists of two half-cycles. The first pass began by placing the tube between the mandrel and the die. Then, the tubes of the initial dimensions were extruded using a cylindrical punch with the same dimensions as the pressed tubes (Figure 1b,c). The second pass began by inverting the die and re-extruding the tube into the channel using a punch, giving it different inner and outer dimensions of 20 and 25 mm, respectively, with a thickness of 2.5 mm. At the end of the second pass, the tube returned to its original diameter (Figure 1d). The imposed strain in each pass was ~2.5, as calculated via Equation (1) [20,25]:(1)ε-PTCAP=2N{∑i=12[2cot(φi2+ψi2)+ψicosec(φi2+ψi2)3]+23lnR2R1 }
where the N, φ, ψ, R_2_, and R_1_ parameters are the number of passes, die channel angle, die angle of curvature, and the tube’s outer and inner radius, respectively (Figure 1a). The maximum imposed strain through the PTCAP after 4 and 2 passes of the pure Cu and brass before tube fracture was 10 and 5, respectively. The shape of the tube before, during, and after the PTCAP process is shown in Figure 1e.

The microstructure observations were performed by different microscopes according to the material processing conditions. A scanning electron microscope (SEM) model FEI INSPECT–S50 (FEI, Austin, TX, USA) was used in the case of the annealed Cu and brass tubes. The microstructure samples were cut perpendicular to the pressing direction at the mid-span tube to 2.5 mm thick, 8 mm in width, and 10 mm long. The Cu and brass annealed samples were then ground, polished, and, finally, etched in a solution of 5 g FeCl_3_, 100 mL ethanol, and 5 mL HCl for 10–40 s [38]. 

Conversely, the microstructure observations of the Cu and brass tubes after PTCAP processing were obtained using a CS corrected-field-emission transmission electron microscopy (TEM) microscope (JEOL JEM-2100F, Tokyo, Japan) operated at 200 KeV. The TEM samples were ground and then mechanically polished with alcohol and diamond paste suspensions on both sides. A flat and shiny surface was obtained via further polishing using colloidal silica and ethanol mixtures for 3 h. Disc-shaped samples with a diameter of 3 mm were punched from the polished material and electro-polished using a solution of 30% HNO_3_ + 70% CH3OH at −35 °C. After the first and final treatments, the samples’ selected area electron diffraction (SAED) patterns were acquired. The average grain sizes of all SEM and TEM microstructure observations were obtained using a linear interpretation method.

The hardness of the Cu and brass tubes was measured on their faces from the inner to the outer radii. The tube face was carefully prepared by grinding and was then polished. The measurements were performed at a load of 500 g and with a dwell time of 15 s, using a hardness tester, model HV−1000 (Huanshi, Guangdong, China). Hardness measurements were performed along six different radii for each sample, with an interspacing of 0.5 mm between measurement locations. The average hardness value was calculated from six different readings in the same position for each radius and was then plotted to show the hardness distribution in each case. Conversely, the average value of the measurements represents the overall hardness value in each case. The deformation inhomogeneity index (H_index_), depending on the hardness measurements, was calculated according to Equation (2) [34,37,39]:(2)$${\;H}_{\mathrm{index}}=\sqrt{\frac{\sum {\left(H_{i}-H_{\mathrm{avg}}\right)}^{2}}{N_{h}}}$$
where H_i_, H_avg_*,* and N_h_ are the hardness value at each measurement point, the average hardness, and the total number of measurements, respectively.

### 2.2. Wear Properties

A dry sliding wear test was carried out using a TNO tribometer (Irvine, CA, USA) with a block-on-ring wear tester (Figure 2). Samples of a block shape sized 2.5 × 8 × 14 mm^3^ were cut from the central zone of the tubes before and after PTCAP, using a wire EDM. The wear samples were ground and polished carefully, then cleaned in an acetone path in an ultrasonic device. The wear test was performed against a steel rotating sliding ring (carrier) with a 73 mm diameter and a hardness of 63 Rc. The test was performed under sliding speeds of 0.76 and 1.15 m/s, with sliding distances of 1375.2 and 2062.8 m, and applied loads of 30−50 N. The wear test was performed at RT. The wear mass loss (g) was determined using a digital balance device with 10^−6^ kg sensitivity. The wear surface morphology and analysis were performed using SEM to investigate the wear mechanism in each case.

## 3. Results and Discussion

### 3.1. Microstructure Characterization

Figure 3a,c shows the SEM photomicrographs of the annealed Cu and brass tubes. An annealed twining grains microstructure with an average grain size of 48.2 and 69.1 μm, with grain-size variation ranges of 14.03−162.5 and 35−162.5 μm, were noted in the case of the Cu and brass tubes, respectively (Figure 3b,d). 

After PTCAP processing, the Cu sample microstructure changed visibly. The microstructure was a combination of elongated grains, a sub-grain structure, and tangled dislocations. The average grain size was 575.4 nm, with a grain size range of 358–878.1 nm (Figure 4a,b). Interestingly, the microstructure features formed after one pass were very close to those noted in the PTCAP of Cu [20]. Furthermore, the SAED pattern of the Cu tube after one pass (Figure 4a) indicates the presence of low-misorientation-angle grain boundaries (LAGBs). After two passes, the microstructure evolved into a mixture of equiaxed and elongated grains with an average grain size and grain-size range of 227.96 nm and 95.9–644.3 nm, respectively (Figure 4c,d). Finally, after four passes, the elongated grains disappeared; the microstructure evolved into an equiaxed grain microstructure, with an average grain size and grain size range of 138.6 and 54.3–227.8 nm, respectively (Figure 4).

Interestingly, the peak value in the grain-size distribution histograms of the different samples was near the average grain size in each case (Figure 4b,d,f). The increase in the imposed strain from 2.5 up to 10 after one and four passes is effective in the evolution of the grains from elongated to equiaxed ultrafine grain (UFG) microstructure. Moreover, increasing the strain was influential in the evolution of the grain boundaries into high misorientation angle grain boundaries (HAGBs), as noted by the SAED pattern (Figure 4e). The SAED pattern of the PTCAP Cu tube processed to 4 passes, as shown in Figure 4e, indicates a pattern that consists of rings with constant intensity and many diffracted beams. Therefore, many small grains with multiple orientations with HAGBs are noted within the selected field of view. Relative to the SAED pattern of the PTCAP Cu tube processed to 1 pass shown in (Figure 4a) with a higher degree of the dispersed diffracted beams that indicate the formation of LAGBs. A similar observation of the transformation of the SAED pattern of UFG microstructure with LAGBs to the SAED pattern of UFG microstructure with HAGBs with increasing the imposed strain was noted during the SPD of Cu, Ni, and Al alloy [40,41,42] as that observed in the current study. Therefore, the PTCAP of copper tubes up to four passes effectively evolved the microstructure into UFG microstructure with HAGBs.

The present results agree with previous studies [20,21] about the effectiveness of the PTCAP in the grain refinement of the Cu tubes to UFG size using a die angle of 120° and pressing rate of 5 mm/min. Furthermore, the present results were similar to those noted through the grain refinement of Cu bulk samples processed by ECAP [34,36,43]. The grain size, grain size range, and even the microstructure features in the current study and those noted in the case of PTCAP and ECAP of Cu [20,21,36,43,44] were close to each other. Interestingly, the grain size and range obtained after 4 passes under an imposed strain of 10 of 138.6 nm and 54.3–227.8 nm were so close to that of 150–300 nm, observed after the PTCAP of Cu [20,21], and that of 70–390 nm of the ECAP of Cu [43,44] obtained under an imposed strain of 9−4. Furthermore, the formation of HAGBs noted after four passes was also noted in ECAPed Cu samples [44]. Moreover, the microstructure evolution of the Cu tubes in the current work was so close to those of Cu processed by cryogenic rolling [45]. Those observations indicate the accuracy of the microstructure observations obtained in the current work.

Interestingly, it was noted that the grain size of the different copper tubes after PTCAP was a mixture of fine and coarse UFG. The presence of large grains is due to the large die angle and low pressing rate used in the current study, which reduces the imposed strain and allows a slight degree of grain growth. Thus, the processing condition in the current study allows obtaining a microstructure with a mixture of different grain sizes. Therefore, the present conditions enhance the strength (hardness) and conserve a reasonable degree of sample ductility, as previously noted [46,47].

The microstructure of the brass tube after one pass consisted mainly of elongated grains with LAGBs (Figure 5a). The elongated grains’ width and length in the brass tube after one pass (Figure 5a) were smaller than those of the Cu tube after one pass (Figure 4a). After one pass, the brass tube’s average grain size and grain size range decreased to 472.3 nm and 338.4–651.8 nm (Figure 5a,b). Then the microstructure of the brass tube transformed into an equiaxed grain microstructure with an apparent decrease in the average grain size and grain size range down to 142.7 nm and 62.9–276 nm with HAGBs (Figure 5c,d) after two passes.

The present observations agree with the decrease in brass grain size noted after the PTCAP of brass tubes [24,25,27] and the ECAP of bulk Cu-40% Zn samples [48,49]. Interestingly, the obtained grain size of the brass tube in the current study was smaller than that of the 1 μm and 300 nm that was noted after the ECAP of brass samples in [48], due to the higher strain of 2.5–5 in the current study. The comparison between the Cu and brass tubes after two passes proves that the brass tubes have a smaller grain size and grain size range than the Cu ones. The smaller grain size with a narrower grain-size range in the brass tubes is due to the brass samples’ greater initial hardness and brittleness relative to copper. The greater brittleness of the brass tubes is also confirmed through limited deformation in the case of the brass tubes, which can only be processed for up to two passes. Although the SPD processing of the Cu contributes to forming a nano- or UFG-grained microstructure, due to the dynamic recovery or recrystallization of the Cu grains Cu is considered a high stacking fault energy (SFE) material. The SPD-processed Cu samples suffer from grain growth. Therefore, adding Zn to copper and other alloying elements (resulting in the formation of brass alloy) decreases the SFE of the copper by suppressing the cross-slip of screw dislocation [49]. The work-hardening capability of the material during SPD is enhanced by suppressing the dynamic recovery to obtain a more refined grain structure (smaller grain size) via accumulating the dislocation in the copper alloy, due to the presence of alloying elements. Therefore, adding elements like Zn, as in the case of brass in the current study, produces PTCAP tubes with a finer grain size than in the copper tubes, as previously confirmed [49].

### 3.2. Mechanical Properties

#### 3.2.1. Hardness Results

Figure 6 shows the effect of the PTCAP processing on the hardness and hardness inhomogeneity index of the Cu and brass tubes. The hardness of the Cu tube was increased visibly from 66.54 to 123.19 HV after the first pass. Then, the hardness was increased with a lower rate and increased up to 135.13, 139.01, and 144.95 HV after processing in two, three, and four passes, respectively (Figure 6a). The increase in hardness at a higher rate after the PTCAP’s first pass was followed by a lower rate of increase when increasing the number of passes; this was also observed in the case of the Cu tubes, bulk, and powder samples when processed by different SPD methods [20,36,49,50,51,52,53]. The apparent increase in hardness after the first pass is due to the higher degree of grain refinement in the first pass; the lower rate of increase corresponds to the slower rate of the grain size decrease. The Cu tubes’ hardness values after PTCAP were equal to or higher than those of Cu deformed by PTCAP, TCEC, and ECAP [20,36,49,50,51,52,54,55,56].

The hardness of the brass tubes has similar behavior, noted in the case of the Cu tubes. The hardness was increased obviously by 146.86% and 18.45% after one and two passes (Figure 6b). The increase of the brass tubes’ hardness is due to the grain size decrease. Therefore, the decrease in grain size contributes to the hardness increase, according to the Hall–Petch relationship [57,58,59], as shown in Equation (3) [60]:(3)$$\mathrm{HV}=H_{0}+k_{H}d^{1/2}$$
where H_0_ and k_H_ are the appropriate constants that are associated with the hardness measurements, respectively.

The hardness of the brass tube when processed with a die angle of 135° and pressing rate of 2 mm/min was equal to or higher than those previously noted after PTCAP treatment of the brass tubes [25,27] and bulk samples processed by CGP [61] (Figure 6b). Furthermore, the observations of the higher hardness of the brass tubes relative to that of the copper tubes agree with the microstructure observation (Figure 3, Figure 4 and Figure 5). As the addition of the Zn increases the hardness and, thereby, the brittleness of the brass, the samples underwent further refinement of the grains; therefore, the hardness increased. 

The hardness distribution before and after the PTCAP of Cu and brass tubes across their thickness has a homogenized distribution (Figure 7 and Figure 8). The hardness varied from 64.49 HV on the inner surface to 67.25 HV on the outer surface, with a hardness difference of 4.23 HV in the case of the annealed Cu tube. Then, the hardness values difference between the inner and outer surfaces decreased to 2.64, 3.43, and 2.24 HV after one, two, and four passes, respectively (Figure 7). Therefore, the hardness distribution after the PTCAP of Cu can be considered a homogenized distribution, with a notable decrease in the hardness difference between the inner and outer surfaces. The results of the decrease in the hardness difference between the inner and outer surfaces agreed with the decrease in the hardness inhomogeneity index (Figure 6). The hardness inhomogeneity index decreased from 1.19 in the annealed Cu tube to 1.07, 0.99, 0.85, and 0.36 after the PTCAP of the Cu tubes at one, two, three, and four passes, respectively. This homogenized distribution can be explained by the tubes’ thinner thickness, which allows homogenized deformation under high imposed strain. Those observations are congruent with those noted after the PTCAP of the Cu tubes [20,21]. Moreover, the hardness measurements were taken through the thickness of the tubes, as the hardness increased through the tube thickness with the increase in the outer radius of the sample, according to Equation (1). Therefore, the hardness values were increased from the inner to the outer radius of the tube. Then, the difference between hardness values in the inner and outer surfaces decreased with the increase in the number of passes (the increase of the imposed strain), as noted regarding the improvement in deformation homogeneity and the hardness distribution maps (Figure 6 and Figure 8).

The brass tubes also have homogenized hardness distribution (Figure 6 and Figure 8), as noted previously in the case of the PTCAP of brass tubes [25,27]. The decrease in the hardness difference between the inner and outer surfaces of the brass tubes after the PTCAP was also combined with a decrease in the hardness inhomogeneity index from 1.33 in the brass annealed tube down to 1.29 and 0.37 after PTCAP of the brass tubes up to 1, and 2 passes. Therefore, the PTCAP process produced homogenized tubes with a higher degree of hardness homogenization distribution than the ECAPed processing because of corner gap formation [62,63].

#### 3.2.2. Wear Properties

##### Wear Mass Loss

Figure 9 shows the influence of load and distance on the wear mass loss of Cu and brass tubes before and after PTCAP processing. The load increased the wear mass loss by 67–88, 32–53, 48–52, and 30–45% in the annealed Cu tube and after PTCAP processing with up to one, two, and four passes under different sliding distances. Moreover, the sliding distance increase from 1375.2 to 2062.8 m increased the wear mass loss by 60–42, 52–37, 50–36, and 50–22% in the case of the annealed Cu tube, and processed in up to one, two, and four passes under different loads. A similar observation was noted for the brass tubes (Figure 9a). Wear mass loss increased by 40–45, 22–33, and 26–28% when increasing the load from 30 to 50 N under different sliding distances in the case of the annealed brass tubes processed in up to one and two passes. Furthermore, the mass loss increased by 44–50, 37–40, and 34–36%, increasing the distance from 1375.2 to 2062.8 m under different applied loads in the annealed tubes processed in one and two passes.

Although the mass loss increased when increasing the load and distance, it reduced along with the number of PTCAP passes. The mass loss decreased by 35% after the PTCAP of Cu and brass tubes in up to two passes and further decreased down to 50% of its value after processing Cu tubes in up to four passes. Therefore, the PTCAP processing enhances the wear resistance of Cu and brass by 50 and 35% after deformation from up to four and two passes. The improvement of the wear resistance after PTCAP is due to the hardness increase (Figure 6). Hardness is considered the main factor in controlling the wear resistance, according to Archard’s law, as indicated in Equation (4) [64]:(4)$$V=K\frac{\mathrm{LN}}{\mathrm{HV}}$$
where V, N, L, K, HV are the volumetric wear loss, the applied load, the total sliding distance, the wear coefficient, and the hardness of the wear surface, respectively. Therefore, the PTCAP technique is efficient in improving the wear resistance of Cu and brass tubes. As the grain is refined, the microstructure transforms into a UFG one and increases the hardness and thereby the wear resistance (in the form of a decrease in wear mass loss). The efficiency of the PTCAP in improving the wear can be further proven by comparing the effect of different SPD processes on the wear rate of copper alloys [34,35] (Figure 9c). The Cu and brass tubes after PTCAP processing have superior wear resistance relative to the Cu alloy and brass samples that were processed with ECAP and CGP [34,35]. The wear mass loss values of the tubes after PTCAP were lower by 8 to 10 times than those for tubes processed with ECAP and CGP [34,35].

It is clear from Figure 9a,b that the brass tubes have higher wear resistance than the Cu tubes. The brass tube wear mass loss was smaller by 65–100% than the Cu tubes before and after PTCAP. Further confirmation of the superior wear resistance of Cu and brass tubes after PTCAP can be observed from the scratch profile, as shown in Figure 10. The wear scratches of the Cu and brass annealed tubes were wider and deeper than those noted after PTCAP. Furthermore, the width and depth of the scratches decreased when increasing the PTCAP number of passes. Moreover, the brass tubes before and after the PTCAP process have narrower and shallower scratches than the Cu ones. These observations confirm the improvement of wear resistance in the PTCAP Cu and brass tubes relative to those of the annealed tubes, with the higher wear resistance of the brass tubes.

##### Worn Surface Morphology and Analysis of Copper and Brass Tubes

Figure 11 shows the worn surface of copper tubes tested before and after PTCAP processing under a distance and a load of 1375.2 m and 50 N, respectively. The wear surface morphology of the Cu annealed tube (Figure 11a) featured severe wear with a high degree of delamination, cracks, and adhesive wear mechanisms. Increasing the PTCAP number of passes decreased the severe wear, as the degree of delamination decreased without the cracks appearing after one pass and two passes (Figure 11b,c). Then the cracks completely disappeared, and the wear mechanism turned into a combination of adhesive and abrasive wear mechanisms after four passes (Figure 11d). The transformation of the wear mechanism after PTCAP is due to the increased hardness of the Cu tubes (Figure 6a). A similar observation of the transformation of the wear mechanism into an abrasive wear mechanism with slight adhesive wear was also noted after the SPD processing of the Cu and Cu alloy samples [34,65].

The EDS of worn Cu tubes before and after the PTCAP indicates the presence of O_2_, as shown in Figure 12. The analysis of the Cu annealed tube’s worn surface indicates the presence of 3.52% of O_2_ (Figure 12a). After PTCAP processing, the percentage of O_2_ decreased slightly down to 3.49 and 3.07% after one and two passes (Figure 12b,c), respectively. Finally, the O_2_ percentage was reduced to approximately half of the copper annealed tube (Figure 12d). Therefore, oxidation can be considered a wear mechanism in the copper tubes before and after PTCAP. The decrease of the O_2_ percentage after PTCAP is due to the increase in hardness, as noted after SPD using different methods [34,65].

Figure 13 shows the morphology of brass tubes before and after the PTCAP. The wear mechanism of the annealed brass tube was determined as delamination and adhesive wear. Furthermore, the worn surface of the annealed brass tube was free from cracks, as noted in the case of the Cu annealed tube. Therefore, the wear degree in the case of the annealed brass tube (Figure 13a) was less than that of the Cu tube (Figure 11a). After PTCAP of up to 2 passes, the wear mechanism of the brass tube changed into an abrasive wear mechanism (Figure 13b). Therefore, the brass tubes after the PTCAP acquired more enhancement of their wear resistance. The present results from enhancing the brass tubes’ wear resistance and the wear mechanism transformation were observed in the brass samples processed by CGP [35].

Analysis of the worn brass tubes before and after PTCAP indicates the presence of O_2_ on the surface of the annealed tube (Figure 13c). However, the EDS of the brass tube after two passes indicates that the O_2_ disappeared (Figure 13d). Therefore, the PTCAP processing of the brass tubes improves their hardness, and oxidation cannot be considered a wear mechanism in the case of the brass tubes after PTCAP. A similar disappearance of O_2_ on the worn surface eliminated oxidation as a wear mechanism, whereas increasing brass hardness was also noted after deformation by CGP [35].

Figure 14 shows the effect of increasing the distance on the morphology and analysis of the worn surface of the Cu and brass tubes processed in up to four and two passes. The increase in distance increased the degree of adhesive wear, as shown and proved in Figure 14a,b. The analysis of the worn Cu samples indicates the presence of 2.63% of O_2_. Therefore, the O_2_ percentage increased with the sliding distance (Figure 14c). However, analysis of the brass tube (Figure 14d) indicated that the sample surface was free from O_2_. Therefore, oxidation cannot be considered a wear mechanism in the case of the brass tubes after PTCAP processing.

## 4. Conclusions

In the current research, it can be concluded that:The PTCAP processing of Cu and brass tubes produced a microstructure with a mixture of fine and coarse grain sizes.A notable increase in Cu and brass tubes’ hardness and wear resistance was obtained after PTCAP processing.PTCAP-processed Cu and brass tubes have wear resistance higher by 50-35% than the annealed tubes and the bulk samples processed with ECAP and CGP in previous works.The tubes’ PTCAP processing obviously influences the wear mechanism and its transformation from one scenario to another.The wear samples’ SEM photomicrographs and EDS patterns prove the wear mass loss and mechanism results.

## Figures and Tables

**Figure 1 materials-15-02985-f001:**
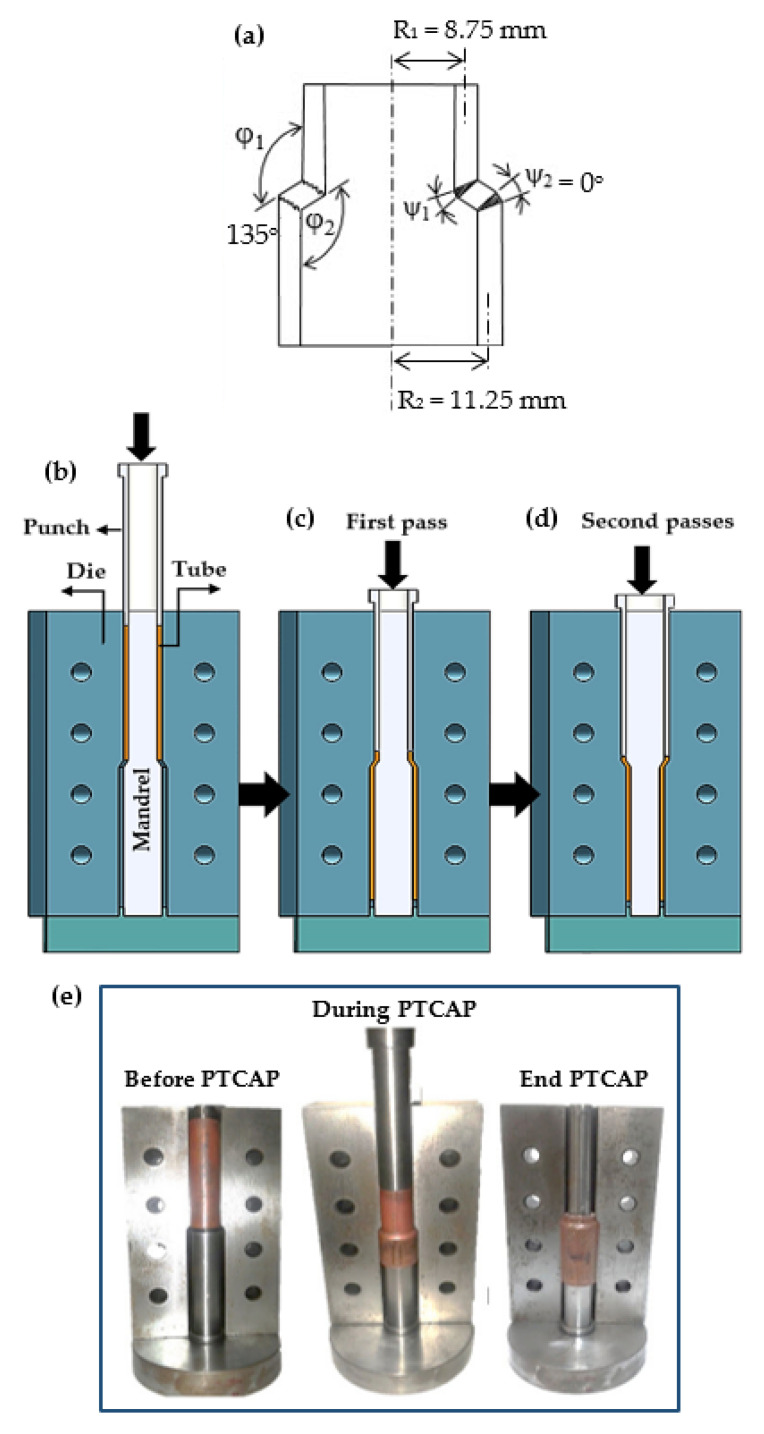
Schematic of the preparation: (**a**) PTCAP die working and geometry, (**b**) preparation setup, (**c**) first half of the cycle, (**d**) second half of the cycle, and (**e**) PTCAP process steps and sample shape.

**Figure 2 materials-15-02985-f002:**
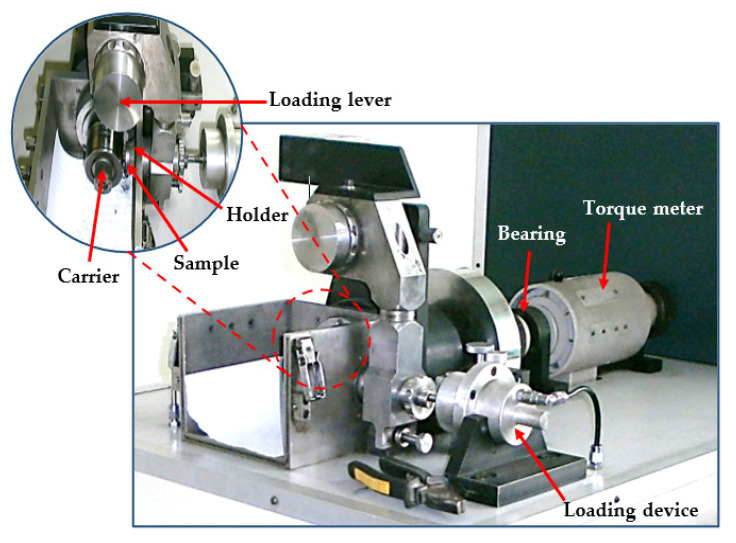
Block-on-ring test rig of the model tribometer.

**Figure 3 materials-15-02985-f003:**
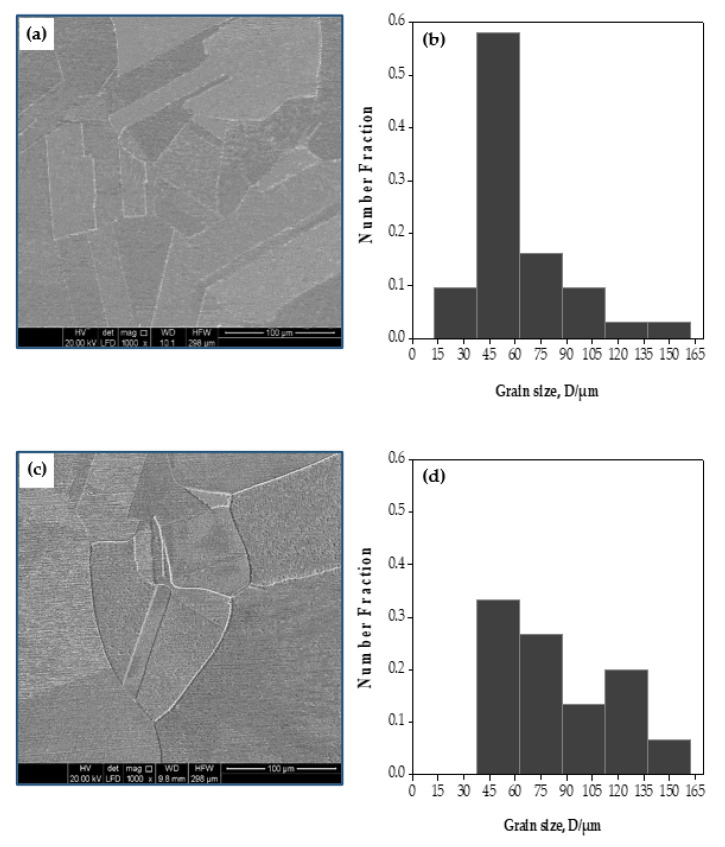
SEM photomicrograph of the microstructure and the corresponding grain size distribution of annealed (**a**,**b**) copper and (**c**,**d**) brass tubes.

**Figure 4 materials-15-02985-f004:**
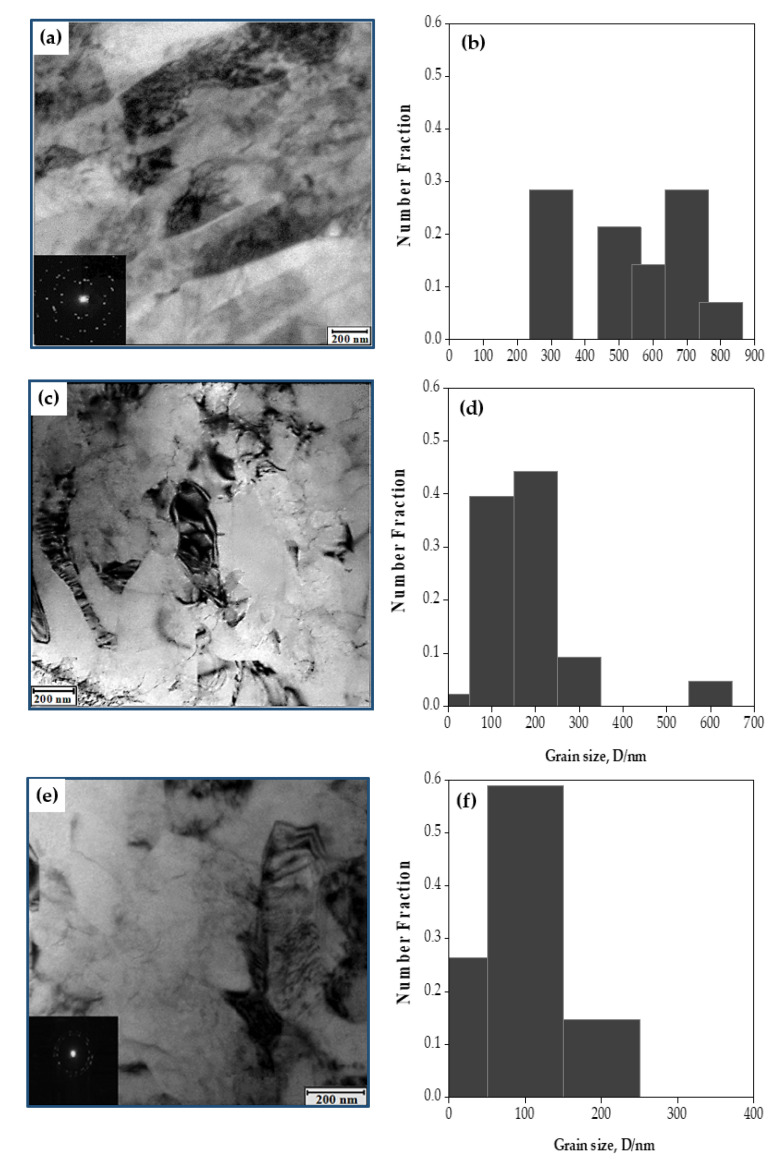
TEM photomicrographs of the microstructure and the corresponding grain-size distribution of copper tubes processed in (**a**,**b**) 1 pass, (**c**,**d**) 2 passes, and (**e**,**f**) 4 passes.

**Figure 5 materials-15-02985-f005:**
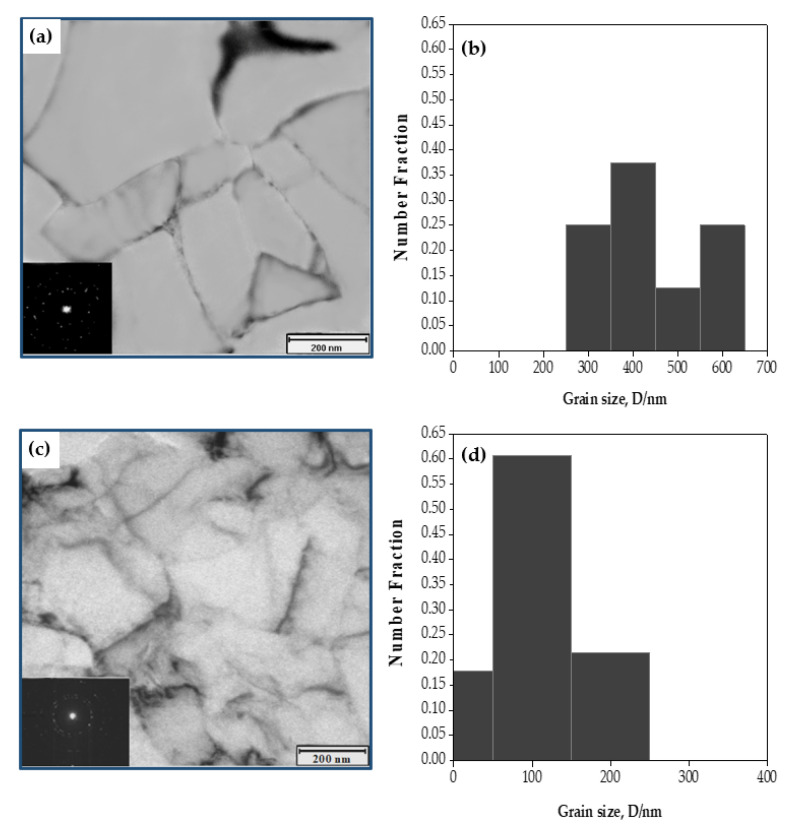
TEM photomicrograph of the microstructure and the corresponding grain size distribution of brass tubes processed in (**a**,**b**) 1 pass and (**c**,**d**) 2 passes.

**Figure 6 materials-15-02985-f006:**
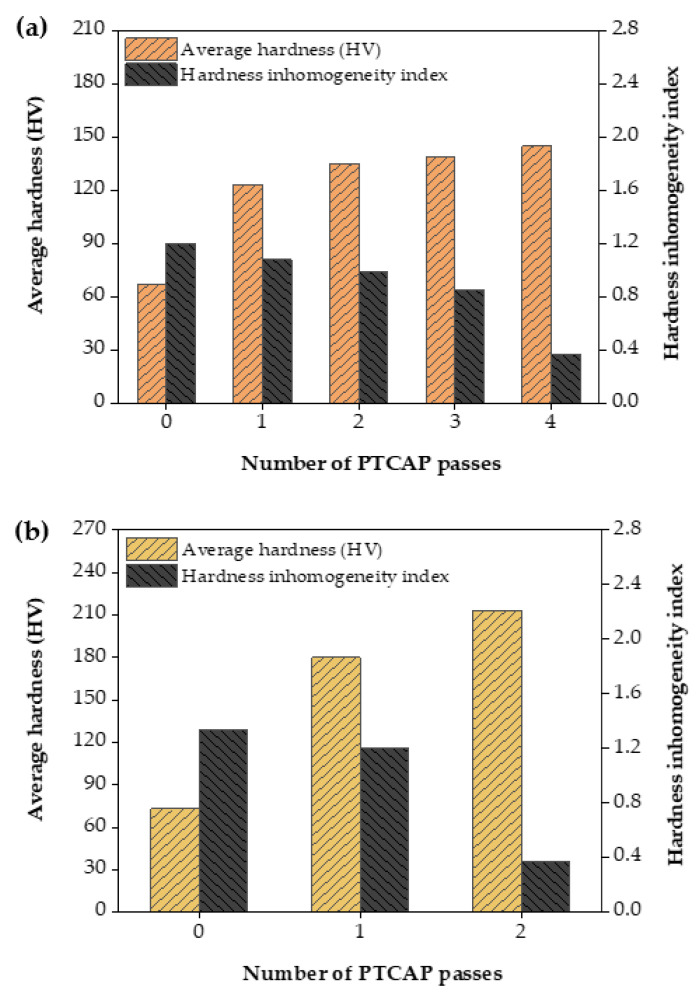
Average hardness and hardness inhomogeneity index before and after PTCAP passes in (**a**) copper and (**b**) brass alloy tubes.

**Figure 7 materials-15-02985-f007:**
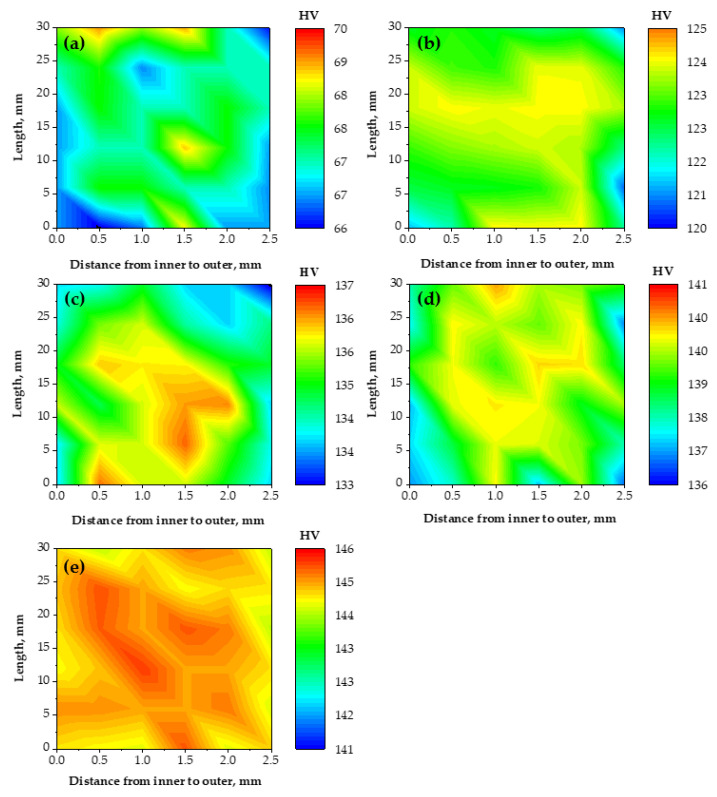
Hardness distribution map of the copper tube along the tube thickness cross-section with distance from the inner to the outer: (**a**) annealed, and after PTCAP processing of (**b**) 1 pass, (**c**) 2 passes, (**d**) 3 passes, (**e**) 4 passes.

**Figure 8 materials-15-02985-f008:**
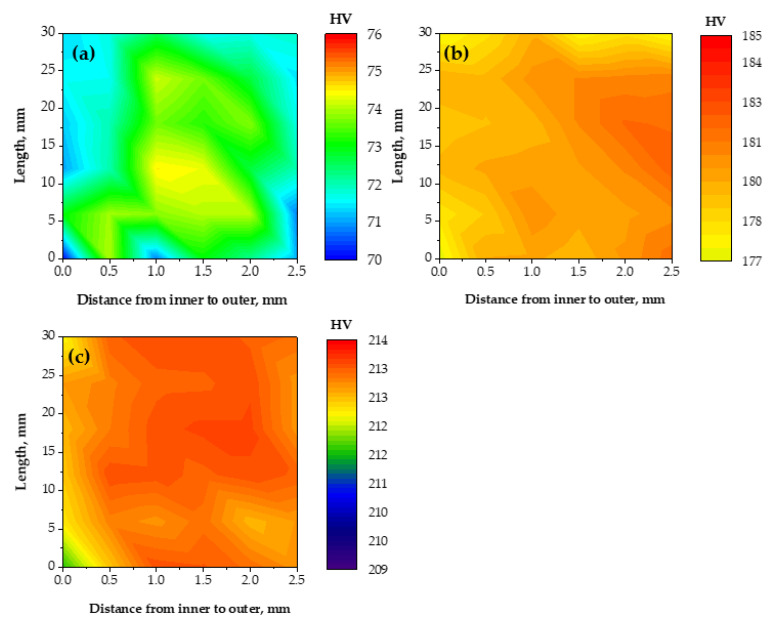
Hardness distribution map of the brass tube alloy along the tube thickness cross-section, with the distance from the inner to the outer surface: (**a**) annealed, (**b**) after the first pass, and (**c**) after the second pass of the PTCAP processing.

**Figure 9 materials-15-02985-f009:**
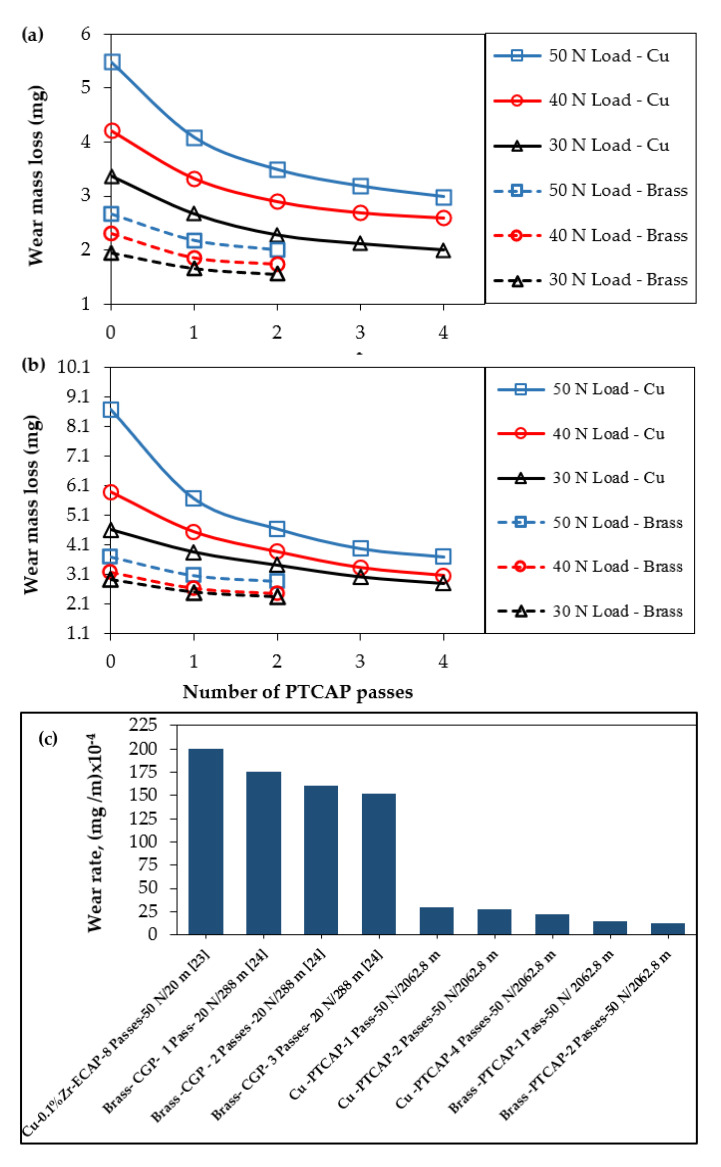
Effect of the number of passes and the applied load on the wear mass loss of the Cu and brass tubes under sliding distances of (**a**) 1375.2, (**b**) 2062.8 m; (**c**) shows a comparison with previous works.

**Figure 10 materials-15-02985-f010:**
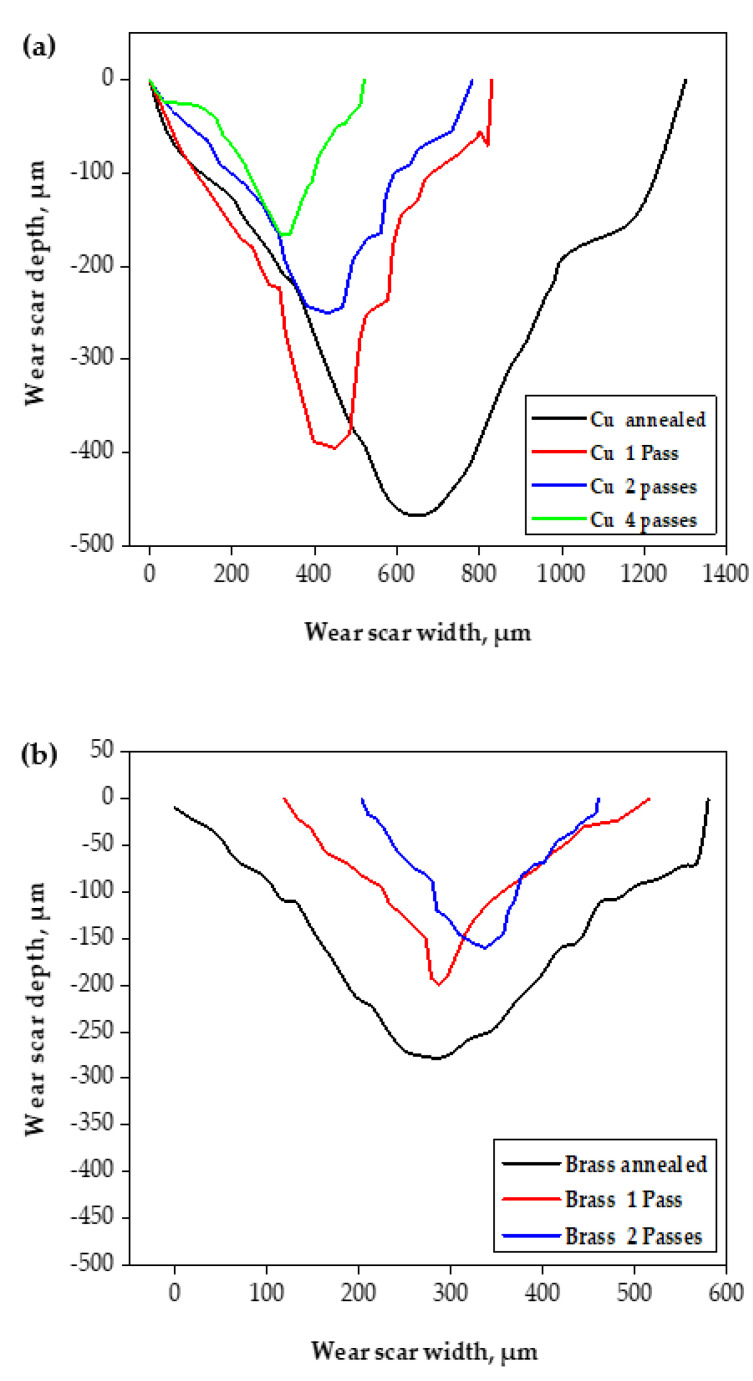
Cross-sectional profiles of the wear scars of the (**a**) Cu and (**b**) brass tubes after the wear test, under an applied load of 50 N and a sliding distance of 1375.2 m.

**Figure 11 materials-15-02985-f011:**
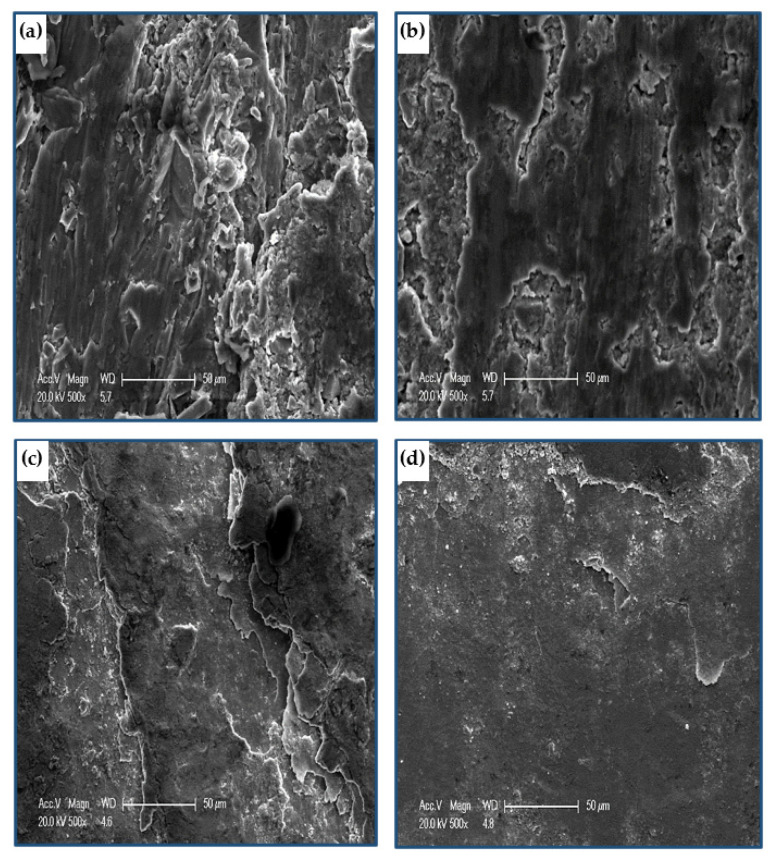
Worn surface morphology of Cu (**a**) annealed, (**b**) 1 pass, (**c**) 2 passes, (**d**) 4 passes under a sliding distance of 1375.2 m, and an applied load of 50 N.

**Figure 12 materials-15-02985-f012:**
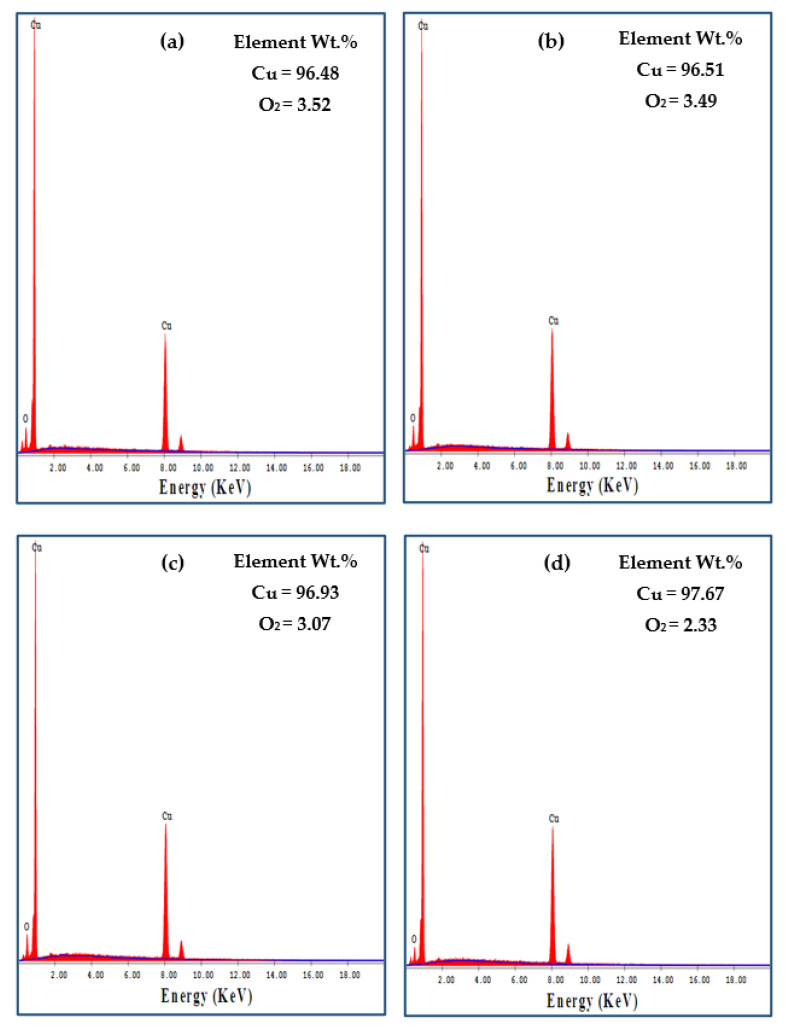
EDS analysis of the worn surface of (**a**) annealed, (**b**) 1 pass, (**c**) 2 passes, and (**d**) 4 passes of Cu tubes under a sliding distance of 1375.2 m and an applied load of 50 N.

**Figure 13 materials-15-02985-f013:**
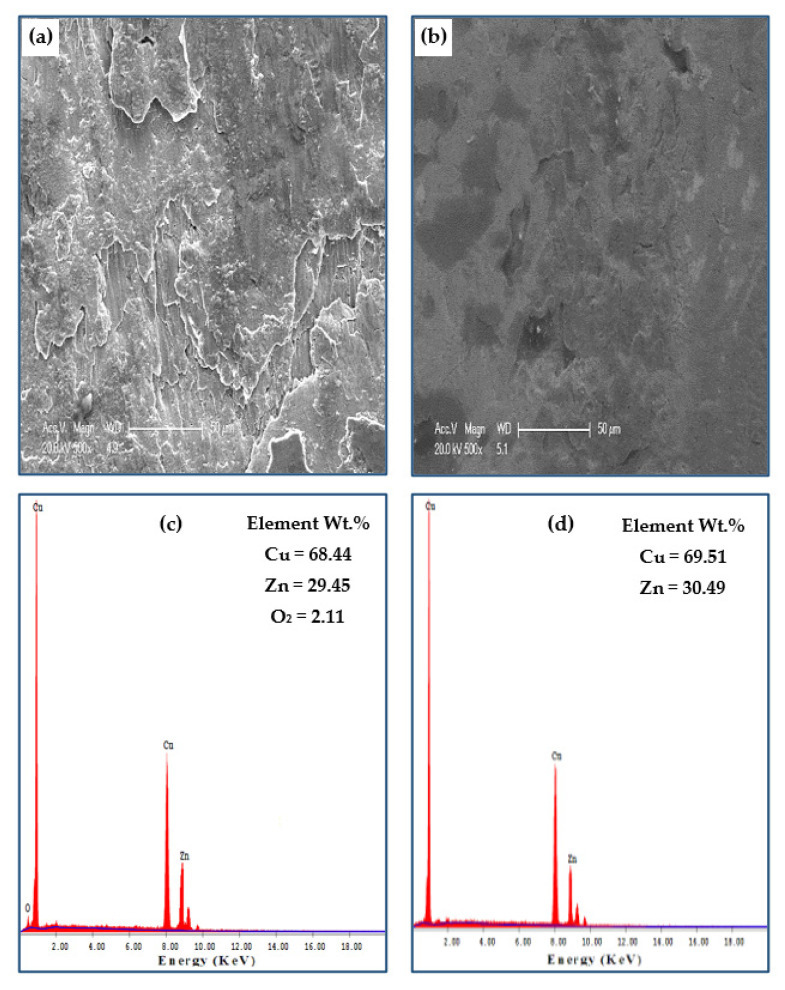
Worn surface morphology and EDS, analysis of brass tubes (**a**,**c**) annealed, (**b**,**d**) 2 passes, under a sliding distance of 1375.2 m, and an applied load of 50 N.

**Figure 14 materials-15-02985-f014:**
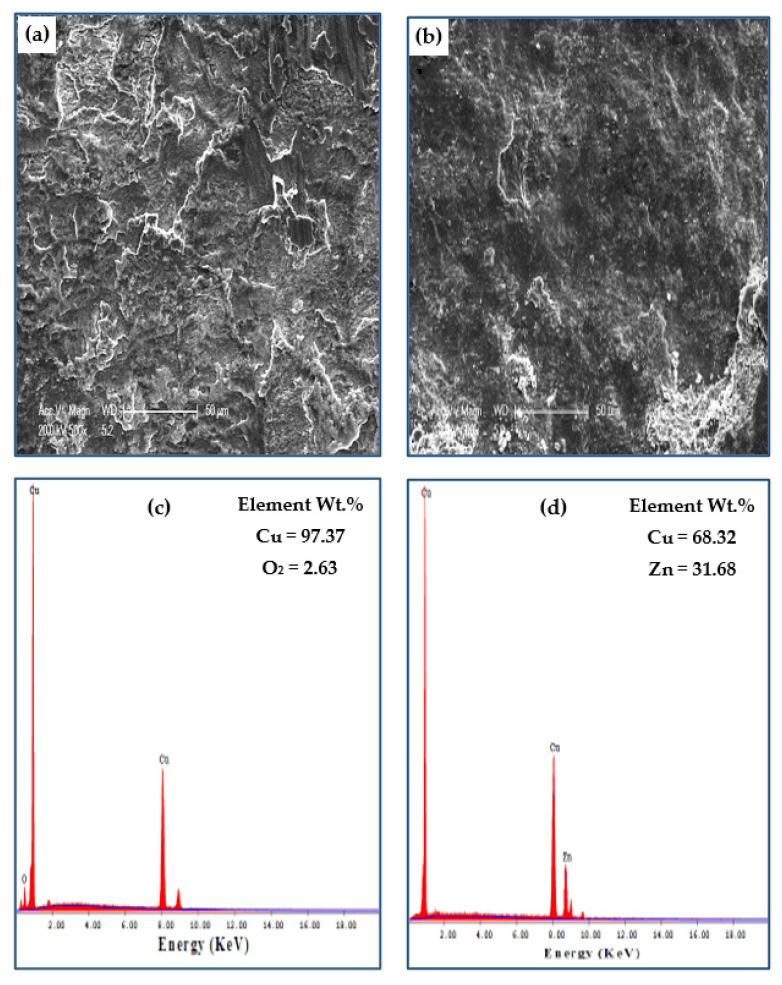
Worn surface morphology and EDS, analysis (**a**–**c**) of the Cu tube after 4 passes and (**b**–**d**) of the brass tube after 2 passes, under a sliding distance of 2062.8 m and an applied load of 50 N.

**Table 1 materials-15-02985-t001:** Chemical composition of the pure Cu and brass tubes (in wt %).

Element (wt %)	Zn	Al	Sn	Pb	Fe	Ni	Mn	Si	P	Be	Mg	As	Cr	S	Co	Cu
Pure Cu	0.0317	0.0001	0.0005	0.0035	0.0001	0.0016	0.0001	0.0005	0.0031	0.0016	0.0075	0.0015	0.0001	0.0037	0.0003	Bal.
Brass	29.85	0.0011	0.0012	0.0135	0.0054	0.00	0.00	0.0001	0.00	0.00	0.00	0.0003	0.00	0.0018	0.00	Bal.

## Data Availability

The data presented in this study are available on request from the corresponding author. The data are not publicly available due to the extremely large size.

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
