# Peer review of "Influence of Parallel Tubular Channel Angular Pressing (PTCAP) Processing on the Microstructure Evolution and Wear Characteristics of Copper and Brass Tubes"

_materials, 2022, doi:10.3390/ma15092985_

Round 1

Reviewer 1 Report

All the review comments are listed in the word document.

Author Response

AUTHOR REPLY TO THE REVEIWER COMMENTS ON materials-1669316

Dear the editor,

Thank you for conveying to me the recommendations of the referee.

We revised our manuscript altered in line with the suggestions of the reviewers. We would like to respond to their comments as follows.

Answer to the Reviewer#1

Thanks for your helpful comments. Please note that all answers about your comments are highlighted by green color throughout the manuscript.

Abstract

(1) The brackets should be deleted in the lines of 20, 22, 24 and 25, and I
think the word PTCAP is necessary here in the text and also it has
been explained before.

Answer:  All of the brackets in lines 20, 22, 24, and 25 were deleted. Moreover the brackets of the abbreviations were revised along the paper.  

(2) The key words should be adjusted. The word “wear characteristics” is
enough, and “wear surface characterization” can be deleted.

Answer:  The key words were adjusted and key word wear surface characterization was deleted. Page 1. Line 27

Introduction

(1) In the line 57, it is better to further explain the reason for considering a microstructure with a mixture of fine and coarse grain.

Answer:  Further explain the reason for considering a microstructure with a mixture of fine and coarse grain was add." Recently, further investigations on using a combination of smaller die angles with a slower pressing rate in producing a microstructure with a mixture of fine and coarse grain brass tubes with high hardness, and wear resistance with good ductility were recently conducted preformed [30]. However, using the same condition in the PTCAP of Cu or other brass tube types did not attempt until now. The presence of a microstructure with a mixture of fine and coarse grain consider a multi-modal that consists of a mixture of submicron and micron sizes, and so it can provide a considerably high level of strength and hardness with conserving a reasonable degree of ductility [31-33] relative to the SPD materials with mono-modal ultrafine grains that increase strength and hardness with sacrificing with the ductility." Page 2. Lines 55-65. New references 31and 32 were added and appear in the list of references Pages 21-22. Line 525-528.  

 (2) Line 66, the punctuation error should be corrected.

Answer:  The punctuation error in line 66 was corrected. Page 2. Line73.

         (3) There are still some grammatical errors that makes sentence meaning difficult to be understand, for example the sentence between lines 69-71.

Answer:  The English language was carefully correct throughout the paper, and attention was paid to sentence between lines 69-71 and corrected to "Study the influence of grain refinement after the PTCAP on the mechanical properties of the Cu and brass tubes, mainly hardness and wear mass loss under different applied load and sliding distances." Page 2. Lines 77-79.

Materials and Methods

(1) Line 88, MoS2 should be MoS2.

Answer: MoS2 was corrected to MoS2. Page 2. Lines 95.

(2) It is better to change the size of table 1 in order to match the page.

Answer: Table 1 in the format of journal. Page 3. Line 113.

Results and Discussion

  • It should be better to give further explanation for the reason of smaller grain size in brass tubes, which is simply mentioned with higher hardness and brittleness by the author. (Lines 214-217)

Answer: Further explanation of smaller grain size in brass tubes relative to the Cu was added" Although the SPD processing of the Cu contributes to forming nano or UFG grained microstructure. However, due to the dynamic recovery or recrystallization of the Cu grains, as Cu considers high stacking fault energy (SFE) material. The SPD processed Cu samples suffer from grain growth. Therefore, adding Zn to copper and other alloying elements (the formation of brass alloy) decreases the SFE of the copper by suppressing the cross-slip of screw dislocation [49]. The work hardening capability during the SPD is enhanced by suppressing the dynamic recovery to obtain a more refined grain structure (smaller grain size) by accumulating the dislocation in the copper alloy due to the presence of alloying elements. Therefore, adding elements like Zn, as in the case of brass in the current study, produces PTCAP tubes with finer-grain size than the Copper tubes, as previously confirmed [49].". Page 6-7. Lines 234-245. New reference 49 was added and appear in the list of references Page 22. Line 563-564.  

(2) In Fig.4 (e), according to the TEM microstructure especially the diffraction pattern, it is not clear enough to identify the ultrafine grains with HAGBs. Therefore, it should be replaced by a more typical one, if possible, microstructure characterization by EBSD can also be used here.

Answer: The quality of Fig.4 (e) was enhanced obviously.  Moreover comparing with the only previous work of PTCAP of the Cu reference [20] use TEM in the microstructure characterization in the present work was in the same quality or even better. This relatively little bad quality is due high imposed strain after the processing up to 4 passes in the current study or 3passes in the previous work [20].

(b)

(a)

The TEM photomicrograph of the microstructure of copper tubes processed up (a) previous work reference 20  and (b) current work pass.

Moreover, the identification of the ultrafine grains with HAGBs using the TEM SAED pattern was explained and supported with previous works "The SAED pattern of the PTCAP Cu tube processed to 4 passes shown in (Figure 4e) indicates a pattern that consists of rings with constant intensity and many diffracted beams. Therefore many small grains with multiple orientations with HAGBs are noted within the selected field of view. Relative to the SAED pattern of the PTCAP Cu tube processed to 1 pass shown in (Figure 4a) with a higher degree of the dispersed diffracted beams that indicate the formation of LAGBs. A similar observation of the transformation of the SAED pattern of UFG microstructure with LAGBs to the SAED pattern of UFG microstructure with HAGBs with increasing the imposed strain was noted during the SPD of Cu, Ni, and Al alloy [40-42] as that observed in the current study. Therefore, the PTCAP of copper tubes up to four passes effectively evolved the microstructure into UFG microstructure with HAGBs." Page 6. Lines185-195. New references were added and appear in the list of references Page 22. Line 544-549.   

(3) It is questionable that why the HV hardness varies more or less along   the longitudinal direction (the ordinates in the figures) since it is a extrusion processing.

Answer: First of all, we would like to thank the reviewer for this question. We agree that along the longitudinal direction (pressing direction of the PTCAP) since it is an extrusion processing that makes HV hardness has no noticeable variation in its value along the pressing direction. However, we want to clarify that the hardness measurements were carried out in the transverse direction of the PTCAP. So for more clarification, the following part was added to the manuscript.

"Moreover, the hardness measurements were taken through the thickness of the tubes, as the imposed increases through the tube thickness with the increase of the outer radius of the sample according to equation (1). Therefore the hardness values were increased from the inner to the outer radius of the tube. Then, the difference between hardness values in the inner and outer decreased with increasing the number of passes (the increase of the imposed strain), as noted through the improvement of the deformation homogeneity and the hardness distribution maps (Figures 6,8). Page 11. Lines301-307.

We appreciate the reviewer's and editor kind and instructive comments and hope that the revisions are satisfactory and that the revised version of the paper will be acceptable for publication in materials

                                                                                                            Yours sincerely,

                                                                                                                 The authors

Reviewer 2 Report

This paper studied the effect of Parallel Tubular Channel Angular Pressing (PTCAP) Processing on the microstructure evolution and wear characteristics of copper and brass tubes. The content of this paper is very similar to a recently published paper by the same author in Materials, ‘Parallel Tubular Channel Angular Pressing (PTCAP) Processing of the Cu-20.7Zn-2Al Tube’. Having read that paper, it is found that the introduction, materials and methods, and conclusions are quite similar in text. The authors should significantly reduce the self-plagiarism in text.

The novelty is low, the research content, i.e. the lower die angles of 135o, the lower pressing rate of 2mm/min, have been mentioned and studied in the previous paper in Materials. The only difference is the different materials used. It would make more sense if this paper could focus on different process conditions, say, at high temperature.

The method used in this paper is basically the same as the previous paper in Materials. Both are experimental methods using SEM, hardness test, wear test etc. Again, it would make more sense if this paper could focus on the modelling parts.

For the above two points I made, the novelty would be there if any of these two point is met.

In the introduction, introduce the previous work Ref 31, mention what is the novelty of this paper compared with Ref 31. It would be helpful to add more Refs about the latest developed SPD processes of bulk samples [10.1016/j.ijmachtools.2019.03.002; 10.1016/j.msea.2016.07.044; 10.1016/j.ijmachtools.2021.103771]

What confused me is that, for the same die geometry, in the previous Materials paper, the imposed effective strain in one pass was 1.49, but in this paper it is said to be 2.5.

The conclusion section is very similar to the conclusions in the previous paper in Materials. It needs to be restructured.

Author Response

AUTHOR REPLY TO THE REVEIWER COMMENTS ON materials-1669316

Dear the editor,

Thank you for conveying to me the recommendations of the referee.

We revised our manuscript altered in line with the suggestions of the reviewers. We would like to respond to their comments as follows.

Answer to the Reviewer#2

Thanks for your helpful comments. Please note that all answers about your comments are highlighted by cyan color throughout the manuscript.

  1. This paper studied the effect of Parallel Tubular Channel Angular Pressing (PTCAP) processing on the microstructure evolution and wear characteristics of copper and brass tubes. The content of this paper is very similar to a recently published paper by the same author in Materials, ‘Parallel Tubular Channel Angular Pressing (PTCAP) Processing of the Cu-20.7Zn-2Al Tube’. Having read that paper, it is found that the introduction, materials and methods, and conclusions are quite similar in text. The authors should significantly reduce the self-plagiarism in text.

     Answer:  The tone of the reviewer questions is around the novelty of the current study and that there is an obvious similarity with the previous work in reference 31 that was renumbered as reference 30 in the current version of the manuscript, and we would like to answer about that as follows:-

  • The degree of the current work similarity with the previous works performed before submitting the work to the Materials was performed using iThenticate program as attached here. It can note that the name of the paper of the previous work, reference 30, is not listed. Moreover, the overall similarity percent did not exceed 20%, and the max similarity percent from previous works did not excessed 2% from one reference. iThenticate check was also performed after the revision the overall similarity percent did not exceed 18%, and the max similarity percent from previous works did not excessed 3% from one reference   So how can it be said that the present work has such self-plagiarism in the text? The reviewer can prove this by using iThenticate program.

The first two pages from the iThenticate check of the paper before submission and after revision are attached in the flowing pages.

  • The present work is concerned with investigating the wear characteristics of the Cu and brass tubes processed by PTCAP, which can be considered the second paper about this topic, and the reviewer can prove that through a search about that.
  • The previous work 30 was interested in microstructure, and mechanical properties, including only wear mass loss after the wear test. We want to clarify that the present work deeply investigates the wear mass loss (wear rate) compared to previous SPD works of Cu and brass. Moreover, the Cu and brass tubes' wear mechanism and wear scars were thoroughly investigated. Finally, the wear mechanism results were confirmed by studying the wear surface morphology and analysis of the Cu and brass tubes. We confirm that the current work is the first to be performed about the wear of the PTCAP tubes.
  • The current work objectives have reconstructed the point out that the work is novel. So we confirm the novelty of the current work that focused on enhancing the PTCAP processed Cu and brass tubes' wear properties. Page 2. Lines73-83
  1. The novelty is low, the research content, i.e. the lower die angles of 135o, the lower pressing rate of 2mm/min, have been mentioned and studied in the previous paper in Materials. The only difference is the different materials used. It would make more sense if this paper could focus on different process conditions, say, at high temperature.

The method used in this paper is basically the same as the previous paper in Materials. Both are experimental methods using SEM, hardness test, wear test etc. Again, it would make more sense if this paper could focus on the modelling parts.

    Answer: The authors appreciate the excellent suggestion of the reviewer about using high temperature and modeling to improve the novelty of the work. However, using high temperature during the SPD processes (in the current work PTCAP) leads to the recovery and grain growth, increasing the grain size and deteriorating the hardness and wear properties. Modeling of the SPD processes was performed widely even for the PTCAP in previous works, references 21 and 37 in the current works. The modeling, especially using FEM simulations, needs a long time to reach the optimum boundary and simulation conditions that need many runs before performing the FEM. Therefore starting new work on FEM of the PTCAP need more time. Finally, again authors appreciate the nice suggestion about new work and already think to perform PTCAP FEM simulation under high temperature with confirming the results experimentally.

  1. In the introduction, introduce the previous work Ref 31, mention what is the novelty of this paper compared with Ref 31. It would be helpful to add more Refs about the latest developed SPD processes of bulk samples [10.1016/j.ijmachtools.2019.03.002; 10.1016/j.msea.2016.07.044; 10.1016/j.ijmachtools.2021.103771]

    Answer: The authors appreciate the reviewer's suggestion about adding new references. The new references were added, as references [17-19].  Mentioned in Page 1. Line 42 and appear in the list of references Page 21. Line 493-499 

  1. What confused me is that, for the same die geometry, in the previous Materials paper, the imposed effective strain in one pass was 1.49, but in this paper it is said to be 2.5.

Answer: We agree with the reviewer about this typing mistake; the imposed effective strain after one pass is 2.5 per pass, as mentioned in this current and previous works references 30. Page 3. Line111

The authors confirm that they contacted the journal editor to fix this erratum typing. Further, here is the calculation for the imposed strain equation.

Where: -

  1. The conclusion section is very similar to the conclusions in the previous paper in Materials. It needs to be restructured.

Answer: 1- The conclusion were restructured as follows: pages 19-20. lines 433-442

In the current research, it can conclude that:

  1. PTCAP processing of Cu and brass tubes produced a microstructure with a mixture of fine and coarse grain sizes.
  2. A remarkable increase in Cu and brass tubes' hardness and wear resistance was obtained after PTCAP.
  3. PTCAPed Cu and brass tubes have high wear resistance by 50−35% than the annealed tubes and bulk samples processed by ECAP and CGP in previous works.
  4. The tube PTCAP processing obviously influences the wear mechanism and its transformation from one case to another.
  5. The wear samples' SEM photomicrographs and EDS patterns prove the wear mass loss and mechanism results.

We appreciate the reviewer's and editor kind and instructive comments and hope that the revisions are satisfactory and that the revised version of the paper will be acceptable for publication in materials

                                                                                                            Yours sincerely,

                                                                                                                 The authors
